# National Facilitators and Barriers to the Implementation of Incentives for Antibiotic Access and Innovation

**DOI:** 10.3390/antibiotics10060749

**Published:** 2021-06-21

**Authors:** Christine Årdal, Yohann Lacotte, Suzanne Edwards, Marie-Cécile Ploy

**Affiliations:** 1Antimicrobial Resistance Centre, Norwegian Institute of Public Health, Postboks 222 Skøyen, 0213 Oslo, Norway; 2University of Limoges, INSERM, CHU Limoges, RESINFIT, U1092, F-87000 Limoges, France; yohann.lacotte@inserm.fr (Y.L.); marie-cecile.ploy@unilim.fr (M.-C.P.); 3Global AMR R&D Hub, 10629 Berlin, Germany; suzanne.edwards@dzif.de

**Keywords:** antibiotic innovation, antibiotic access, medicine shortages

## Abstract

Prominent reports have assessed the challenges to antibiotic innovation and recommended implementing “pull” incentives, i.e., mechanisms that give increased and predictable revenues for important, marketed antibiotics. We set out to understand countries’ perceptions of these recommendations, through frank and anonymous dialogue. In 2019 and 2020, we performed in-depth interviews with national policymakers and antibiotic resistance experts in 13 countries (ten European countries and three non-European) for a total of 73 individuals in 27 separate interviews. Interviewees expressed high-level support for antibiotic incentives in 11 of 13 countries. There is recognition that new economic incentives are needed to maintain a reliable supply to essential antibiotics. However, most countries are uncertain which incentives may be appropriate for their country, which antibiotics should be included, how to implement incentives, and how much it will cost. There is a preference for a multinational incentive, so long as it is independent of national pricing, procurement, and reimbursement processes. Nine countries indicated a preference for a model that ensures access to both existing and new antibiotics, with the highest priority for existing antibiotics. Twelve of thirteen countries indicated that shortages of existing antibiotics is a serious problem. Since countries are skeptical about the public health value of many recently approved antibiotics, there is a mismatch regarding revenue expectations between policymakers and antibiotic innovators. This paper presents important considerations for the design and implementation of antibiotic pull mechanisms. We also propose a multinational model that appears to match the needs of both countries and innovators.

## 1. Background

Antibiotic resistance is a serious public health threat, accounting for 33,000 European deaths (in 2015) and 35,000 American deaths annually [1,2]. Bacteria’s ability to become resistant to antibiotics is an evolutionary process that will always exist, but careful interventions limit progression considerably [3]. Slowing antibiotic resistance requires effective infection prevention and control measures as well as ensuring that patients receive the right treatment at the right time at the right dose. Unnecessary use of antibiotics hastens the development and spread of resistance [3]. Lack of access to essential antibiotics contributes to the deaths of about one million children every year [4].

As antibiotic resistance emerges, new and effective treatments are needed. Yet, antibiotic innovation is struggling. Physicians use new antibiotics as a last resort in order to preserve their efficacy. Whereas this is sound stewardship, it dis-incentivizes innovation since unit sales determine revenues. As of September 2020 there were 31 antibiotics and 27 non-traditional antibacterial agents in clinical development for the World Health Organization’s (WHO) Priority Pathogens (excluding those targeting tuberculosis), compared to over one thousand for cancer therapy [5,6]. The preclinical pipeline is stronger with 251 antibacterial candidates [5]. Yet, the scientific risk of failure is higher in earlier development phases [7]. The low revenues and high risks have resulted in large pharmaceutical companies abandoning antibacterial research and development, with only three large companies performing clinical development [5]. Thus, only small companies, dependent upon external financing for survival, remain to research and develop new antibiotics.

Several reports have examined how to stimulate antibiotic innovation [8,9]. Their recommendations include increasing research and development funding (so-called “push” measures) as well as improving the market revenues (so-called “pull” measures). Recommended pull mechanisms delink antibiotic revenues from sales, thereby paying companies for making important new antibiotics available.

In the last five years, the response with “push” funding has been positive, with the formation of the publicly and philanthropically funded organizations Combating Antibiotic-Resistant Bacteria Biopharmaceutical Accelerator (CARB-X) and The Global Antibiotic Research and Development Partnership (GARDP), and the privately funded Replenishing and Enabling the Pipeline for Anti-Infective Resistance (REPAIR) Fund and Antimicrobial Resistance (AMR) Action Fund. These come in addition to the ongoing contributions of the Joint Programming Initiative on Antimicrobial Resistance (JPIAMR), European Union’s Innovative Medicines Initiative (IMI), the United States’ Biomedical Advanced Research and Development Authority (BARDA), and more. Additionally, Germany under its leadership of the G20 Forum supported the formation of the Global AMR R&D Hub, with a mandate to address challenges and improve coordination and collaboration in global AMR R&D using a One Health approach. GARDP acts also as a pull mechanism since it will ensure the availability of its antibiotics in low and middle-income countries.

In response to the calls for pull mechanisms, three countries have implemented or are trialing new reimbursement mechanisms [10]:

England will pay an annual fixed payment determined through a health technology assessment (including both patient and societal value) for the supply of a new antibiotic. The payment is delinked, that is, not dependent upon sales volumes. The pilot has selected two antibiotics. Target implementation date is Spring 2022.

Germany has revised the way it assesses new “reserve” antibiotics, allowing for higher unit prices in line with the value of the new antibiotic.

Sweden has entered into an annual revenue guarantee with suppliers of five newly marketed antibiotics. All antibiotics and suppliers that met the requirements were accepted to enter into an agreement. Swedish hospitals continue to purchase as normal with the funding from the pilot study paying the difference between the guarantee and actual sales. The revenue guarantee amount is delinked from antibiotic sales. The agreements started July 2020 and will continue for two years.

The Pharmaceutical Strategy for Europe (2020) states that the EU will pilot a pull incentive in 2021 [11], and legislation containing a pull incentive was introduced into the American Congress in December 2020 [12] and is expected to be reintroduced in 2021. These initiatives hold promise, but unfortunately while waiting for implementation small innovators are struggling. Three companies with marketed antibiotics went bankrupt in the last two years [13].

At the same time that innovation is struggling, shortages of older antibiotics are increasing [14]. Due to antibiotic resistance patterns and prescribing habits, the markets of some essential antibiotics are small, including those for children. Tendering processes based solely on price and automatic price reductions for generic medicines reduce profitability. This has led some manufacturers to stop producing generic antibiotics, and thereby increasing reliance on few manufacturers. The dependency on sole manufacturers may come as a surprise, when suddenly an antibiotic is no longer available. For example, in 2017 a fire at an active pharmaceutical ingredient factory in China resulted in a global shortage of piperacillin/tazobactam [15]. During the COVID-19 pandemic, supply chains have been unable to meet demand as well as challenged by supply disruptions due to lockdowns and border closures [16].

As a part of the EU Joint Action on Antimicrobial Resistance and Healthcare-Associated Infections (EU-JAMRAI) we set out to understand countries’ perceptions regarding antibiotic incentives, through frank and anonymous dialogue. The aim was to understand the barriers and facilitators of implementing incentives for antibiotic access and innovation.

## 2. Methods

In this study, we performed in-depth interviews with human health policymakers and AMR experts in ten European countries: Belgium, Denmark, France, Germany, Luxembourg, the Netherlands, Norway, Romania, Spain, and Sweden. These insights were made more globally representative by the facilitation of the Global AMR R&D Hub with the inclusion of a further three countries from other continents (Canada, Japan, and South Africa). The European countries were selected from participating countries within EU-JAMRAI with the intention of ensuring representativeness from different geographies and population sizes. The non-European countries were selected as a sample of large countries in different income classifications. We attempted to secure interviews with additional non-European countries, but as this coincided with the COVID-19 pandemic, we were unsuccessful in interviewing policymakers in two additional countries. All interviews were performed in 2019 and 2020.

We used a standard interview guide (Appendix A). To encourage open and inclusive dialogue, all interviews were held under Chatham House Rule, meaning that “participants are free to use the information received, but neither the identity nor the affiliation of the speakers, nor that of any other participant, may be revealed.”

## 3. Results

In total, we interviewed 73 individuals in 27 separate interviews. The barriers and facilitators to implementing incentives for antibiotic access and innovation as stated by national policymakers and AMR experts are summarized in Table 1.

### 3.1. Perceptions Regarding Pull Incentives

Interviewees expressed high-level and general support for antibiotic incentives in 11 of 13 countries. There is a general recognition by all interviewed countries that new economic incentives are needed to maintain a reliable supply to essential antibiotics. However, there was little depth of understanding, with less than half of the countries familiar with the literature on antibiotic incentives. With all of the activity regarding new entities providing push funding, policymakers were often confused regarding the roles and differentiation of the new actors like CARB-X and GARDP.

Countries were also uncertain which incentives may be appropriate for their country, which antibiotics should be included, how to implement incentives, and how much it will cost. The majority of countries are waiting for clear and concise recommendations from a respected entity like the European Commission or the Global AMR R&D Hub, utilizing evidence from the pull mechanisms implemented in Germany, Sweden, and the United Kingdom. Eleven of the thirteen countries interviewed would prefer a multinational incentive, that is, one where countries may opt in, so long as it is independent from national health technology assessment, medicine pricing, and reimbursement processes, which are complex and heterogeneous. There was little interest in a new incentive that would disrupt these national processes, especially since new antibiotics are expected to be used rarely.

### 3.2. A Need for Access to Old and New Antibiotics

Whereas almost all countries stated a concern about the lack of antibiotic innovation, this was not the principal driver to support new incentives. Rather, nine of the eleven countries supporting new incentives indicated a preference for a model that ensures access to both old and new antibiotics, with the highest priority for older antibiotics. Indeed, countries do not have predictable access to generic antibiotics. Twelve of thirteen countries indicated that shortages of existing antibiotics are a serious problem nationally. Eight out of thirteen indicated that this resulted in greater use of broad-spectrum antibiotics, thereby potentially increasing antibiotic resistance. Moreover, with a recurrent unavailability of some antibiotics, doctors change prescribing habits, potentially away from evidence-based prescribing guidelines. Existing antibiotics with fragile availability mentioned in interviews included ampicillin, benzylpenicillin, benzathine penicillin, cefotaxime, cloxacillin, nitrofurantoin, phenoxymethylpenicillin, temocillin, and trimethoprim.

Eight countries indicated that companies recently decided to stop marketing an essential, existing antibiotic in their country. Three countries managed to reverse this decision by awarding higher unit prices. One country secured the commitment of a small company willing to produce an older antibiotic for a higher unit price and assisted the company in the transfer of the marketing authorization. Yet, in some countries, important older antibiotics have never been registered. One country successfully engaged a manufacturer to market an older antibiotic never previously marketed by offering mutual recognition of existing regulatory dossiers. The manufacturer responded that no countries were interested in this older antibiotic. Only later, through ad hoc communications, the country learned that several countries also wanted access to this older antibiotic. These examples of successfully securing access are the exception. In most instances, countries lost access to the antibiotic.

Unpredictable access is not only a challenge for older antibiotics but also for new ones, which are often not widely available, even in high-income countries. Only six of the thirteen countries were aware that the availability of new antibiotics, especially those manufactured by small producers, may be delayed in their country, and these six all represented smaller market countries. Larger market countries were generally unaware that, despite their size, they may be considered an unattractive market.

Regarding supply chain transparency, in most cases countries were unaware and surprised that factory information is generally considered a business secret and is not shared, even between national regulators. One asked “*why isn’t it public, after all we can see where our meat is produced?*” Countries were concerned that older antibiotics are reliant on few manufacturers, particularly active pharmaceutical ingredient manufacturers. Countries expressed interest in further pursing transparency of regulatory dossier information.

### 3.3. Public Health Value of New Antibiotics

Countries were skeptical regarding the public health value of many recently approved antibiotics. All antibiotics that have received regulatory approval in the past decade have been approved with non-inferiority designed clinical trials. That is, the new antibiotic is found to be *not* inferior to a comparator (often generic) antibiotic. These trials do not provide direct clinical evidence of efficacy against multi-drug resistant bacteria. Instead, regulators rely upon data regarding improved spectrum from both in vitro and in preclinical animal infection models. There are several reasons for this clinical trial design, particularly including patients infected with multi-drug resistant bacteria in a trial showing superiority will be too long and costly, and there is no standard treatment allowing an early inclusion and randomization [17]. Policymakers are frustrated that new antibiotics show no greater benefit than existing antibiotics. As one policymaker said, “*Antibiotics are being approved for indications where there is no intention that they will be used. This sends the wrong signal…would prefer that antibiotics are tested against drug-resistance instead. If the trials need to be done in [high-resistance countries] and they are performed according to existing standards, this is preferable.*” Policymakers were clear that incentives should only apply to antibiotics that meet public health needs, i.e., either those on antibiotic prescribing guidelines or new antibiotics that show benefit in clinical situations for unmet public health needs.

Almost all countries agreed that the WHO’s Priority Pathogen List represents their unmet public health needs. One country stated that they agree with the overall list but national needs differ in priority order. Another country, due to significant national divergences, is in the process of establishing its own priority pathogen list. Similarly to the desire for a multinational mechanism, there is also a wish for an independent body to recommend new antibiotics for eligibility based upon an assessment of public health value. European countries anticipate the European Medicines Agency will perform this role. Non-European countries also look to their national regulators.

### 3.4. Incentive Cost

Countries were often concerned about the estimated price tag of potential incentives, as the literature has estimated global revenue amounts in the billions needed to invigorate innovation [8,9]. Many policymakers expressed frustration regarding the lack of engagement from large multinational pharmaceutical companies, given that the revenues of other (often high-priced) medicines are dependent upon effective antibiotics. Most countries are uninterested in estimates of revenues needed to stimulate innovation. Rather, they are interested in paying amounts commensurate with the national value of having the particular antibiotic accessible. The majority of countries were uninterested in incentives that dramatically increase antibiotic prices, as the German model may do, often referring to WHO’s Fair Pricing Forums [18].

## 4. Discussion

Through interviews with policymakers and AMR experts in thirteen countries we have explored the facilitators and barriers to implementing incentives to promote antibiotic access and innovation (Table 1). Country representatives expressed general support for antibiotic incentives with a number of caveats, but are uncertain about incentive design and cost. The main aim of any incentive should be to secure access to antibiotics of public health importance, including existing antibiotics. This means new antibiotics will need to more clearly demonstrate public health value through clinical evidence but this could come from smaller trials or post-approval trials [19]. Almost all countries would prefer a multinational incentive, but one that is independent from national regulatory, pricing, procurement, and reimbursement processes, as these are national responsibilities.

An adjusted version of the Swedish model could meet these criteria. Based upon its resistance surveillance, Sweden estimated its medical need against specific resistant infections four years into the future. This medical need was multiplied by the antibiotic’s package price and then multiplied by 150%, to ensure that companies were rewarded for availability. This calculation resulted in Sweden guaranteeing annual revenues of SEK four million (about €400,000) for newly marketed antibiotics meeting specified requirements [20]. Producers are guaranteed an annual revenue, with the difference between the guarantee and actual annual sales paid through the new incentive. If sales exceed the guarantee amount, the innovator keeps the additional revenues as well as receiving a 10% bonus so long as all contractual conditions have been met. Sweden has entered into two-year contracts with the antibiotic producers and has included national access and stewardship provisions.

The principles of the Swedish model could be extended to a multinational incentive, with the aim that countries could participate in a joint tender with a common contract template (excluding the guarantee amount). Each country would negotiate a separate revenue guarantee with the producer. The process is visualized in Figure 1. This pull incentive would allow countries to delink antibiotic revenues from sales volumes allowing innovators to have greater revenue predictability.

Firstly, eligible antibiotics would be selected either by a country requesting their inclusion or based upon regulators’ recommendations. As most countries require open tendering processes, these calls for tender would likely need to specify eligible antibiotic characteristics (e.g., antibiotics approved for infections with multi-drug resistant bacteria, antibiotics approved against specified Gram-negative pathogens, narrow-spectrum antibiotics for treatment of pneumococcal infections) [20,21].

Once eligibility requirements have been defined, a joint tender would be developed, with a contract template including national access and stewardship requirements. A suggested revenue guarantee calculation would be included. The guarantee amount would be negotiable, allowing for increased revenue guarantees for important new antibiotics developed with private financing.

Participation would always be voluntary, on behalf of both national governments and producers. However, if a company opts in, it must commit to perform good faith negotiations with all interested countries. Once the tender participants are agreed, each country would negotiate individually with the producer and ultimately enter into a contract. Global medicines distributors (like the WHO/GARDP SECURE initiative that plans to accelerate and expand access to essential antibiotics in low and middle-income countries) should also be invited to participate. All of the abovementioned process could be coordinated by a multinational organization like the European Commission or the Global AMR R&D Hub.

This model could also be adjusted where the European Commission (or another entity) could commit to the annual revenue guarantee on behalf of all countries. This could also be independent of national procurement, pricing, and reimbursement processes. For example, if four or more countries requested the Commission to enter into a guarantee for a specific antibiotic, the Commission would negotiate an agreement, including access stipulations, with the company. The difference between the guarantee and the total annual sales would be paid by the participating countries, according to a pre-agreed distribution, such as by gross domestic product.

The aim of this model is to make it easy for countries to implement a pull incentive ensuring sustainable access to antibiotics while at the same time providing market certainty to antibiotic producers. Some may argue the revenues generated through these agreements will be insufficient to provide an attractive market for innovators. Calls to stimulate antibiotic innovation have primarily centered on the needs of antibiotic innovators with price tags of over USD one billion per antibiotic globally paid out over several years [8,9]. Yet, through our model the revenue guarantee amount would be negotiable. National unit prices may also be informed by a health technology assessment, which may include societal value (e.g., the value of reducing transmission of a multi-drug resistant pathogen) in addition to patient value. Both the United Kingdom and Norway are trialing including societal value [22,23]. Additionally, with better clinical evidence, the national unit prices per antibiotic should increase.

It was clear through our interviews that there is currently a mismatch between national policymakers and innovators regarding the perceived value of specific, new antibiotics. This incentive offers an opportunity for countries and innovators to find the middle ground, while giving predictability in both demand and supply.

## 5. Conclusions

During 2020 the world has been challenged by the COVID-19 pandemic. Whereas antibiotic resistance does not emerge or kill as rapidly, it continues to be a serious problem for global public health. We welcome the EU Pharmaceutical Strategy committing to implement a pull incentive for antibiotics in 2021. This paper offers relevant findings to inform this incentive and proposes a model matching national government and innovator needs.

## Figures and Tables

**Figure 1 antibiotics-10-00749-f001:**
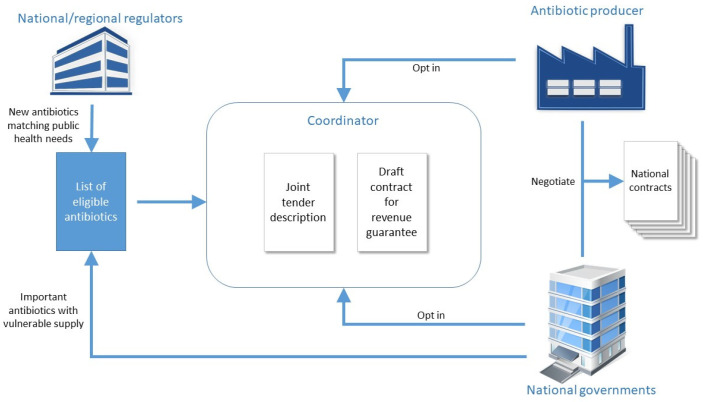
Potential multinational pull incentive based upon principles of the Swedish incentive.

**Table 1 antibiotics-10-00749-t001:** Facilitators and barriers to implementing incentives for antibiotic access and innovation.

Facilitators	Barriers
Countries generally agree that new economic incentives are needed to maintain a reliable supply of essential antibiotics; Evidence regarding the effectiveness and operational cost of implementing different types of pull incentives is being generated in Germany, Sweden, and the United Kingdom; Pull incentives can ensure access to both old and new antibiotics, which is desirable since predictable access to existing antibiotics is a serious challenge in many countries; Pull incentives can be designed to only reward antibiotics that meet public health needs; Almost all countries agree that the WHO’s Priority Pathogen List represents their unmet public health needs for antibiotic innovation; The EU has committed to trial a pull incentive in 2021.	Most countries are uncertain which incentives may be appropriate for their country, which antibiotics should be included, how to implement incentives, and how much it will cost; Most countries prefer a multinational incentive and are waiting for a first mover to organize the process; Countries are skeptical about the public health value of many recently approved antibiotics, which have all been approved through non-inferiority clinical trials; There is a mismatch between the estimated price tag of new pull incentives and the public health value.

## Data Availability

As data was gathered under the Chatham House rule, the individual interview data are unavailable.

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
