# Peer review of "National Facilitators and Barriers to the Implementation of Incentives for Antibiotic Access and Innovation"

_antibiotics, 2021, doi:10.3390/antibiotics10060749_

Round 1

Reviewer 1 Report

Title: Fine.

Abstract:

It is missing the year in which you carried out the study.

Background:

  • Line 82-87: the citation is missing
  • Considering the importance of the topic, I recommend to investigate the topic of antibiotic resistance by reading the following article

Kocuria kristinae: an emerging pathogen in medical practice”

Napolitani M., Troiano G., Bedogni C., Messina G., Nante N., J. Medical Microbiology, 68, 1596-1603, 2019.

Method:

  • The study period is missing.
  • Why did you choose these countries? Are there particular motivations such as high use of antibiotics?

Results:

  • I would describe the sample of respondents, for example I would say the average age, percentage of men and women, etc.
  • I recommend describing the results better, perhaps even summarizing them in a table to better evince the outcomes.

Discussion: fine

Conclusions: fine

References: The bibliography is poor of similar scientific works

Author Response

Dear Dr. Cui and Peer Reviewers,

Thank you very much for your insightful feedback regarding our article, “National Facilitators and Barriers to the Implementation of Incentives for Antibiotic Access and Innovation”. We have made the requested changes. Please find a response to the comments below to each point.

Best regards,

Christine Årdal

Reviewer 1

  1. Abstract: It is missing the year in which you carried out the study.

Response: Thank you, we have incorporated the years in the abstract (line 15).

  1. Background: Line 82-87: the citation is missing.

Response: Reference 10 applies to all three country’s new reimbursement mechanism, including these lines. We hope that this is apparent in the article.

  1. Considering the importance of the topic, I recommend to investigate the topic of antibiotic resistance by reading the following article: “Kocuria kristinae: an emerging pathogen in medical practice” Napolitani M., Troiano G., Bedogni C., Messina G., Nante N., J. Medical Microbiology, 68, 1596-1603, 2019.

Response: Many thanks for drawing our attention to this important emerging pathogen. For the purposes of incentives for antibiotic access and innovation, we rely upon the World Health Organization’s Priority Pathogen List, the current international standard defining the bacteria of which new antibiotics are urgently needed.

  1. Method: The study period is missing. Why did you choose these countries? Are there particular motivations such as high use of antibiotics?

Response: Thank you, we have incorporated the years and rationale for country selection in the Methods section (lines 116-122).

  1. Results: I would describe the sample of respondents, for example I would say the average age, percentage of men and women, etc. I recommend describing the results better, perhaps even summarizing them in a table to better evince the outcomes.

Response: We would normally describe the demographics of the respondent population, as you describe, if this was a sample of individuals from a targeted group. However, it was the countries that selected the appropriate participants based upon their job responsibilities given the content of the interview guide template. Therefore, we believe that age and gender are not pertinent to the results. We have summarized the results section in Table 1 and hope that this clearly conveys the results.

  1. Discussion: fine; Conclusions: fine

Response: Thank you.

  1. References: The bibliography is poor of similar scientific works

Response: Antibiotic incentives for access and innovation is a small field within the overall larger thematic area of antibiotic resistance. We appreciate that the bibliography is not as varied as it could be, but this is the result of a small group of academics focused within this field. We believe that the references represent the applicable citations. 

Reviewer 2 Report

In the manuscript submitted by Ardal et al., the authors provide a relevant and comprehensive paper focused on the challenges related to antibiotic access and innovation.

The authors discussed the incentives mechanisms implemented in the last years and also performed an interesting analysis with national policymakers and antibiotic resistance experts in 13 countries in order to evaluate and understand countries’ perceptions, barriers and facilitators regarding antibiotic incentives.

The authors propose a new potential multinational model that could be useful in order to simplify the implementation of the pull incentive mechanism maintaining a sustainable access and providing marketing certainty to manufacturers.

The paper is well written and shows originality and novelty. The methodology used is rationale, the work is well organized and the quality of the results and the investigation performed is high.

On the basis of these considerations, I think that the quality of this paper is high and is appropriate for this journal. The manuscript is ready to be published in this form.

Author Response

Dear Dr. Cui and Peer Reviewers,

Thank you very much for your insightful feedback regarding our article, “National Facilitators and Barriers to the Implementation of Incentives for Antibiotic Access and Innovation”. We have made the requested changes. Please find a response to the comments below to each point.

Best regards,

Christine Årdal

In the manuscript submitted by Ardal et al., the authors provide a relevant and comprehensive paper focused on the challenges related to antibiotic access and innovation.

The authors discussed the incentives mechanisms implemented in the last years and also performed an interesting analysis with national policymakers and antibiotic resistance experts in 13 countries in order to evaluate and understand countries’ perceptions, barriers and facilitators regarding antibiotic incentives.

The authors propose a new potential multinational model that could be useful in order to simplify the implementation of the pull incentive mechanism maintaining a sustainable access and providing marketing certainty to manufacturers.

The paper is well written and shows originality and novelty. The methodology used is rationale, the work is well organized and the quality of the results and the investigation performed is high.

On the basis of these considerations, I think that the quality of this paper is high and is appropriate for this journal. The manuscript is ready to be published in this form.

Response: Many thanks for taking the time to review this article and for your positive feedback. We appreciate your time and efforts.

Reviewer 3 Report

The problem of antibiotic resistance is an important problem of contemporary medicine. Incentives helping antibiotic development and availability have been suggested as a measure to combat resistance.

The paper is dealing with several experts opinion coming from several countries within and outside Europe, about antibiotic pull mechanisms. It also suggests a model matching the innovator needs and the needs of national governments. 

I have no important notes to make, I think the paper is well written and satisfactory from the language point of view. It is an interesting and original work to read.

Author Response

Dear Dr. Cui and Peer Reviewers,

Thank you very much for your insightful feedback regarding our article, “National Facilitators and Barriers to the Implementation of Incentives for Antibiotic Access and Innovation”. We have made the requested changes. Please find a response to the comments below to each point.

Best regards,

Christine Årdal

The problem of antibiotic resistance is an important problem of contemporary medicine. Incentives helping antibiotic development and availability have been suggested as a measure to combat resistance.

The paper is dealing with several experts opinion coming from several countries within and outside Europe, about antibiotic pull mechanisms. It also suggests a model matching the innovator needs and the needs of national governments.

I have no important notes to make, I think the paper is well written and satisfactory from the language point of view. It is an interesting and original work to read.

Response: Many thanks for taking the time to review this article and for your positive feedback. We appreciate your time and efforts.